# Ketoacid Analogues Supplementation in Chronic Kidney Disease and Future Perspectives

**DOI:** 10.3390/nu11092071

**Published:** 2019-09-03

**Authors:** Laetitia Koppe, Mariana Cassani de Oliveira, Denis Fouque

**Affiliations:** 1Department of Nephrology, Hospices Civils de Lyon, Centre Hospitalier Lyon-Sud, F-69495 Pierre-Bénite, France; denis.fouque@chu-lyon.fr; 2Univ Lyon, CarMeN Laboratory, INSA-Lyon, Inserm U1060, INRA, Université Claude Bernard Lyon 1, F-69621 Villeurbanne, France; 3Department of Medical Clinic, Botucatu Medical School, Universidade Estadual Paulista—UNESP, Botucatu 18618-687, Brazil; maricassani@gmail.com

**Keywords:** chronic kidney disease, low protein diet, ketoacid analogues, intestinal microbiota, dialysis

## Abstract

Diet is a key component of care during chronic kidney disease (CKD). Nutritional interventions, and, specifically, a restricted protein diet has been under debate for decades. In order to reduce the risk of nutritional disorders in very-low protein diets (VLDP), supplementation by nitrogen-free ketoacid analogues (KAs) have been proposed. The aim of this review is to summarize the potential effects of this dietary therapy on renal function, uremic toxins levels, and nutritional and metabolic parameters and propose future directions. The purpose of this paper is also to select all experimental and randomized clinical studies (RCTs) that have compared VLDP + KA to normal diet or/and low protein diet (LPD). We reviewed the SCOPUS, WEB of SCIENCES, CENTRAL, and PUBMED databases from their inception to 1 January, 2019. Following duplicate removal and application of exclusion criteria, 23 RCTs and 12 experimental studies were included. LPD/VLPD + KAs appear nutritionally safe even if how muscle protein metabolism adapts to an LPD/VLPD + KAs is still largely unknown. VLPD + KAs seem to reduce uremic toxins production but the impact on intestinal microbiota remains unexplored. All studies observed a reduction of acidosis, phosphorus, and possibly sodium intake, while still providing adequate calcium intake. The impact of this diet on carbohydrate and bone parameters are only preliminary and need to be confirmed with RCTs. The Modification of Diet in Renal Disease study, the largest RCTs, failed to demonstrate a benefit in the primary outcome of the decline rate for the glomerular filtration rate. However, the design of this study was challenged and data were subsequently reanalyzed. However, when adherent patients were selected, with a rapid rate of progression and a long-term follow up, more recent meta-analysis and RCTs suggest that these diets can reduce the loss of the glomerular filtration rate in addition to the beneficial effects of renin-angiotensin-aldosterone system (RAAS) inhibitors. The current evidence suggests that KAs supplemented LPD diets should be included as part of the clinical recommendations for both the nutritional prevention and metabolic management of CKD. More research is needed to examine the effectiveness of KAs especially on uremic toxins. A reflection about the dose and composition of the KAs supplement, the cost-effective features, and their indication to reduce the frequency of dialysis needs to be completed.

## 1. Introduction

End-stage kidney disease (ESKD) is a condition associated with a high mortality and poor quality of life combined with extremely high costs. Using interventions for delaying the need to start a kidney replacement treatment is, therefore, a major challenge. Experimentally, Brenner et al. [1] showed that high protein intake induced marked kidney hypertrophy, which is an increase in glomerular pressure and hyperfiltration that negatively impacts kidney function. Chronic kidney disease (CKD) is characterized by the accumulation of a number of organic solutes called uremic toxins. Many of these uremic toxins are produced by the degradation of dietary amino acids by intestinal microbiota and appears to accelerate CKD progression. Based on these observations, a reduction in protein intake can be expected to preserve renal function and reduce uremic toxicity. The main limitation of this diet is the risk of malnutrition and cachexia. 

Different dietary protein regimens have been tested: low–protein diets (LPD, 0.6 g protein/kg/day) or very low–protein diets (VLPD: 0.3–0.4 g protein/kg/day) supplemented with essential amino acids (EAAs) or nitrogen-free ketoacid analogues (KAs). KAs are precursors of corresponding amino acids since they can undergo a transamination, e.g., a chemical reaction that transfers an amino group to a ketoacid to form a new amino acid (Figure 1). This pathway is responsible for the deamination of most amino acids. Through this conversion, KAs can be utilized in place of their respective EAAs without providing nitrogen products while re-using available nitrogen already in excess during CKD. If a diet does not provide enough EAAs or calories, then the nitrogen balance can become negative and could partly induce cachexia. Therefore, administration of KAs has been proposed to improve protein status while limiting the nitrogen burden on the body. VLDP + KAs are likely also efficient because the calcium content of KA preparation could allow a better correction of mineral metabolism impairment. Different compositions of KAs and EAAs have been tested, with most of them containing four KAs (of the EAA isoleucine, leucine, phenylalanine, and valine), one hydroxyacid (of the EAA methionine), and four amino acids considered essential in CKD (tryptophan, threonine, histidine, and tyrosine) (Table 1).

Since the publication of the perspective by Shah et al. [2] on the role of KA supplementation in the management of CKD, great effort was undertaken to perform randomized controlled trials (RCTs). This review outlines the potential mechanisms of action and efficacy of KAs in renal function and metabolic parameters with a particular emphasis on potential anabolic effects and reduction of uremic toxins production. Our review mainly focuses on recent experimental data and RCTs. Lastly, we propose new fields of investigation for KAs in future research.

## 2. Methods 

We performed a literature search of trials in SCOPUS, WEB of SCIENCES, CENTRAL, and PUBMED databases from their inception to 1 January, 2019 initially without a language restriction. The search strategy used the terms renal or ESRD or end stage renal disease or kidney or CKD or chronic kidney disease and ketoacids or keto analogs or very low protein diet. Titles and abstracts of publications returned from the search were screened for relevance by three authors (L.K., M.C.d.O., and D.F). Clinical studies were included if (i) full text was available in English, (ii) focused on a randomized controlled trial, and (iii) investigated the effect of KAs in non-dialysis CKD patients of any age. Abstracts, case reports, narrative reviews, editorials, letters, and practice guidelines were excluded. We identified 13,132 potentially relevant references from the search (12,453 records without duplicates, Appendix A). After title and abstract screening, 140 full-text articles were considered for inclusion. Twenty-three full-text studies were selected in order to analyze the potential effects of KAs. For experimental studies, we performed a literature search on the PubMed database from inception to 1 January, 2019. We only kept experimental studies, which used a control group. 

## 3. Potential Benefit of Ketoacid Analogues 

Do we have evidence in CKD of specific KAs actions on the reduction of kidney disease-associated comorbidity? New emerging studies suggest that restricted VLDP + KAs may improve renal function and nutritional status, while preventing hyperparathyroidism, insulin resistance, and accumulation of uremic retention solutes (URS), as summarized in Figure 2. The main concern about the interpretation of the literature is the fact that KAs are not given solely but in association with other EAAs and under LPD/VLPD condition. In particular, we do not know if a supplementation of KA alone without low protein diets has any benefit on metabolic disturbances related to CKD. Few studies [3,4,5,6,7] compared KAs supplementation with the same protein restriction and it is difficult to decipher if “KAs effects” are solely the consequence of a decrease of protein intake or if they act specifically. Another interrogation is the reproducibility of the diet composition in different groups. The composition of fibers, acid load, or sodium is difficult to assess and frequently not specified in dietary surveys, which can influence the results. In order to have a more detailed picture of the effects of KAs during CKD, the main experimental trials and RCTs have been summarized in Table 2 and Table 3. 

### 3.1. Anabolic Action of Ketoacid Analogues 

CKD is associated with muscular wasting and there are some concerns about potential catabolic and cachectic effects of LPD/VLPD. Muscle is an adaptive tissue that responds to diet, exercise, and hormones, which may impact protein metabolism. KAs in general and ketoleucine, in particular, have been shown to reduce muscle protein degradation [39]. However, how muscle protein metabolism adapts to a LPD/VLPD + KAs in CKD is still an open question. Leucine kinetics studies have shown that adaptation to dietary protein restriction involves a reduction in leucine flux and oxidation, which leads to a more efficient use of dietary amino acids, and postprandial inhibition of protein degradation with a reduced ureagenesis. These results were similar in CKD patients with an LPD [40] and with a VLPD + KAs [39,41,42]. This reduction in amino acid oxidation during VLPD + KAs persisted during a 16-month follow-up [43].

In addition, in animal models, KAs supplementation may play a protective role on muscle atrophy [8,11,12]. In particular, in 5/6th nephrectomy rats, an LPD + KAs compared to an LPD alone was able to suppress ubiquitin-proteasome system activation and protected skeletal muscle from atrophy and from oxidative damage. KAs can prevent a decrease in the activity of the mitochondrial electron transport chain complex and increase mitochondrial respiration [8,12]. LPD + KAs decreases autophagy markers in muscle, but there was no difference in inflammation in skeletal muscle [11]. It should be noted that the anabolic effect of KAs can be partially explained by the reduction of acid load associated with the reduction in protein intake.

### 3.2. Role of Ketoacid Analogues in Intestinal Metabolism and Uremic Toxins Production 

The uremic syndrome is attributed to the progressive retention of a large number of compounds called uremic retention solutes (URS). Protein bound uremic toxins derived from gut microbiota have emerged as a major class of URS. High levels of indoxyl sulfate (IS) (indol metabolites), p-cresyl sulfate (PCS) (p-cresol metabolite), and trimethylamine-N-oxide (TMAO) have been associated with an increase in cardiovascular risk and renal disease progression [44]. CKD as well as food intake have a deep influence on gut microbiota composition with an increase of pathobionts [45] and recent data have highlighted that modulation of intestinal microbiota appears to be an attractive strategy to reduce URS production [44]. There are some interrogations about the fact that KAs contained a rather large amount of tryptophan/tyrosine, which may have generated more URS. One clinical trial reported that a VLPD + KAs reduced IS serum levels in non-dialyzed CKD patients [23]. The reduction of URS may be explained by the reduction of protein intake and vegetarian diet since Marzocco et al. observed that the higher protein intake (observed during LPD compared to VLPD) was the major determinant of IS levels [23]. In addition, Teplan et al. were able to see a decrease in asymmetric dimethylarginine (another URS) with the same protein intake [4]. Until now, no large RCT has been performed to test the effectiveness of KAs strategy at reducing URS and this is eagerly needed. 

The impact of intestinal function and intestinal microbiota with the presence of LPD/VLPD + KAs is unknown. Previous data suggest that the intestinal absorption of KAs is unaffected by CKD [46]. Recently, using modern mass spectrometry in germ-free mice, the serum concentration of BCAAs was shown to be reduced whereas their fecal concentration was increased. This suggests that these amino acids are less absorbed by the colonic mucosa but are intensely metabolized by the colonic microbiota instead. Therefore, the impact of KAs supplementation on gut microbiota in a uremic condition must be considered and explored. 

One constant effect of VLPD/LPD + KAs treatment is the reduction of serum urea. Initially, it was thought that an important quantity of urea was hydrolyzed during CKD and the amino-group released from urea could aminate KAs and form new AAs. Studies indicate that the reduction in net urea generation with LPD + KA is due to decreased amino acid degradation [41,42]. The view that urea is simply a marker of renal dysfunction has been challenged by recent publications [47]. The microbial enzyme urease issued from the gut microbiota hydrolyzes urea and locally produces excessive amounts of ammonia [48]. Urea and ammonia in the intestinal gut could influence the gut microbiota composition, URS production, and disrupt the intestinal barrier, which is involved in the pathogenesis of inflammation in CKD [48]. At disease-relevant concentrations, urea induces reactive oxygen species (ROS) production and causes insulin resistance [49] and beta-cell dysfunction [47] by modifying insulin signaling molecules by O-linked β-*N*-acetylglucosamine. In addition, urea is able to contribute to posttranslational modification of proteins via the breakdown product cyanate through a process called protein carbamylation, which promotes atherosclerosis and mortality during CKD [50]. In good agreement, in a recent RCT including 60 CKD patients, the VLPD + KAs compared to a moderate protein diet (0.7 g/kg/day) significantly decrease serum urea levels and was associated with a reduced protein carbamylation [20].

### 3.3. Other Roles of Ketoacid Analogues

KA supplements have been associated with an improved bone metabolism, insulin sensitivity, and a decrease in blood pressure. Serum bicarbonate also increased. However, the reduction in protein intake and vegetarian diet induce a similar phenotype [51]. Therefore, the specific role of KAs on these parameters is not clear and more studies are needed.

In a large observational study performed by Bellizzi et al. [52], patients who received LDP + KAs had better blood pressure. In fact, there was a correlation between salt intake and blood pressure since salt intake is reduced by the LPD. In a recent RCT, Milovanova et al. reported that LPD + KAs improve blood pressure when compared to the control group with a similar salt and protein intake (0.6 g/kg/day) [3]. These observed hemodynamic effects could be due to the reduction in salt intake and possibly to an additional vasodilating effect of BCAAs-KAs supplementation.

KAs are synthesized as a calcium salt and daily provide an amount of 600 mg of calcium for a person of 60 kg. Therefore, calcium salt may exert a direct alkaline effect and phosphorus binding. This effect is likely amplified by a vegetarian diet. It is well-known that fruits and vegetable are an effective intervention to correct metabolic acidosis [53,54]. However, Gennari et al. reported no effect of diet on serum total CO_2_ in patients receiving a VLPD + KAs. The increase in serum bicarbonate was related to the decrease in protein intake [27]. However, in a recent RCT with a comparable protein intake, KAs were able to directly increase serum bicarbonate [3]. VLPD + KAs induce a reduction in serum phosphorus only one week after initiation of treatment and this was sufficient to reduce serum FGF23 levels [22]. In a longer six-month study, VLPD + KAs was efficient to improve mineral metabolism [20]. Lineadeau et al. [37] observed an improvement in osteo-fibrotic as well as osteo-malacic changes on bone biopsies after 12 months of treatment with Kas. This fact is independent of calcium intake. Indeed, the control group was supplemented in a comparable amount of calcium than the KAs group. By contrast, Bernhard et al. [6] observed no difference on calcium, phosphorus, and serum bicarbonate in a study where all groups receive calcium and bicarbonate supplementation.

Disorders of glucose homeostasis affect approximately 50% of patients suffering from CKD [55,56], which play a major role in mortality [57]. Pilot clinical studies demonstrated a potential impact of VLPD + KAs compared to an LPD on insulin sensitivity in severe CKD patients [58,59]. However, an LPD is able to improve insulin sensitivity and it is difficult to decipher if KAs have an additive effect [60]. More recently, Teplan et al. have highlighted that, with a similar protein intake, KAs supplementation was able to decrease glycated hemoglobin and visceral body fat, and improve lipid metabolism [4].

Lastly, Di Iorio et al. reported that a VLPD+ KAs diet allowed for a long-lasting reduction of the erythropoietin (EPO) dose required to maintain serum hemoglobin. In this study, the variation of EPO dose directly correlated with the improvement in parathyroid hormone levels. This suggests that the effect of KAs on anemia control was related to the control of phosphorus metabolism [31].

We strongly suggest that such research should be continued to better understand the mechanisms involved during KAs diets. Better knowledge on the interaction between protein intake, gut microbiota, and URS production offers new ways to analyze the mechanisms of action of KAs and increase the motivation of physicians, dietitians, and patients for implementing this diet.

## 4. Do Ketoacid Analogues Have an Impact on Renal Function and Mortality in CKD?

The beneficial role of KAs supplementation on renal function is well established in experimental models, but the effects are different in human clinical trials. LPD/VLPD + KAs were able to decrease proteinuria, renal fibrosis, and severe glomerular sclerosis in different CKD rodent models, as disclosed in Table 2 [9,10,13,14,15,17,18,19]. However, the majority of these preclinical studies have compared a normal protein diet vs. VLPD/LPD + KAs. It is, therefore, difficult to decipher the specific effect of the KAs supplementation [9,10,15,17,18,19]. Gao et al. highlighted that supplementation of KAs increased the Kruppel-like factor-15, which is a transcription factor shown to reduce fibrosis in 5/6 nephrectomized rats [14]. In this study, rats under LPD + KA had 1% less protein in their diet than rats under LPD but this was likely not sufficient to explain the improvement in renal function. To demonstrate specific effects of KAs, Zhang et al. [10] reported that mesangial cells treated with serum derived from 3/4 nephrectomized rats rapidly expressed higher expression of Angiotensin II and Angiotensin I receptors compared to a serum obtained from sham rats. Serum from CKD rats fed LPD significantly inhibited these abnormalities and this effect was even more pronounced with serum from CKD rats fed with LPD + KAs, which suggests that KAs may be directly involved or respond to the greater decrease in protein intake. This is a fact that limited the deleterious activation of renin-angiotensin system.

The interpretation of literature about the clinical efficacy of KAs seems controversial and depends on the chosen inclusion criteria. A recent meta-analysis (*n* = 10 RCT, 1010 patients in CKD stage 4) in non-diabetic CKD patients including the large positive study from Garneata et al. [21] and Milovanova et al. [3] showed that VLPD reduced the number of patients reaching ESRD [61]. Another recent meta-analysis with 661 patients, VLDP + KA significantly reversed the decrease of eGFR (mean difference = 2.74, 95% confidence interval = (0.73, 4.75), *p* = 0.008) in CKD patients [62]. It should be noted that, in the majority of these studies, protein intake was different between groups and the selection of studies is questionable, which makes it difficult to interpret the true role of KAs.

The vast majority of RCTs testing VLPD + KAs reported a decrease in proteinuria. In the large MDRD study, patients with proteinuria >3 g/24 h were more protected against CKD progression with VLPD + KAs [35]. Garneata et al. did not observe a beneficial effect on proteinuria of a VLPD + KAs. However, proteinuria was low in that particular study (<1 g/24 h) [21]. By contrast, in patients with a proteinuria >1 g/24 h, a VLPD + KAs was able to significantly decrease proteinuria [20,52,63,64]. It could be suggested that the anti-proteinuric effect of KAs is more important in patients with a higher proteinuria.

Until now, there is no evidence that LPD/VLPD + KAs have an impact on patients’ survival. A post hoc analysis of the MDRD study demonstrated a slightly increased mortality rate in the VLPD + KAs group after 10 years of study completion (1.95: 95% CI; 1.15–3.20) [25]. However, in this study, the diet adherence was not followed after the termination of the study, and there was no information on the clinical survey and treatments of the patients. The Bellizzi’ historical cohort with approximately 200 patients followed up for more than 10 years did not show adverse effects and suggested possible greater survival [65]. Mortality of patients consuming VLPD + KAs was 8% patients/year, while, in the control group, it was 10% [65]. In another long-term observational study (>10 year), Chauveau et al. did not observe a correlation between the death rate and duration of diet [66]. In recent RCTs, no difference in death or cardiovascular events was observed [3,21]. It should be noted that these studies were not designed to answer this question and larger RCTs are recommended.

## 5. Is a Keto Acid Analogues Supplementation a Modern Treatment?

There is some evidence about potential benefits of LPD/VLPD + KAs on renal outcomes but the effects seem small and this questions its generalization compared to modern treatments such as renin-angiotensin-aldosterone system (RAAS) inhibitors. In a recent review [67], we reported that there was no clinical evidence for synergistic effects of the RAAS inhibitor and LPD. Experimental and clinical studies evaluating the additive effect of an LPD and RAAS inhibition are few, of low quality and limited duration, and only demonstrate significant beneficial effects on proteinuria. The recent study by Garneata et al. in which 71% of participants were receiving an RAAS inhibitory treatment, showed that the probability of worsening kidney function or need for dialysis was lower in the VLPD + KAs group when adjusted for RAAS inhibitors (hazard ratio, 0.10, 95% CI, 0.05-0.20) [21]. Bellizi et al. studied 45% of patients who were under RAAS inhibition and observed a 35% decrease in proteinuria under an LPD and KA combination compared to a normal diet (vs. −1% in the free diet group) [65]. However, in two recent RCTs, the use of RAAS inhibitors were not detailed or have low incidence [3,20].

The other major concern is a potential risk of malnutrition and kidney cachexia. The majority of studies confirmed the safety of VLPD + KAs. In experimental models, the addition of Kas to the LPD largely prevented the loss of weight and completely normalized serum albumin levels [13,14]. In the MDRD study, patients with protein restriction lost about 2 kg and showed a decrease in other anthropometric parameters during the first four months only, likely because of reduced energy intake. Biochemical and anthropometric indices of nutritional status were generally well within normal limits during follow-up and at the end of the MDRD study [35]. In all RCTs testing VLPD + KAs therapy with dietary management performed by trained dietitians reported no change in serum albumin [5,21,26,29,30,33,68], which is subjective to the global assessment score [21,68] or BMI [33]. In recent meta-analyses, Hahn et al. [61] have reported that, among four studies comparing nutritional measures with VLPD or LPD (*n* = 291), the mean final body weight was 1.4 kg higher with VLPD when compared to LPD. Therefore, the risk for wasting during VLPD is 0.6%. This is a magnitude that is not different from normal diets (0.4%). However, the specific role of Kas was not tested in this study [61]. Using whole-body DEXA in an observational study, patients with VLPD + KAs showed a significant decrease in lean tissue mass and an increase in total body fat. These different modifications occurred abruptly during the first three months, and then stabilized or slightly improved thereafter [69]. In another cohort, after at least two years, muscle strength increased in CKD diabetic patients under VLPD + KAs [70].

The routine use of VLPD + KAs in clinical practice remains challenging. Their benefits were mainly seen in patients who showed good compliance to an LPD. The study of Garneata et al. was the first study confirming this fact [21]. Only 14% of the screened patients were kept for the study and randomized, which underlines the limitation to propose this therapy to a larger number of patients. Similarly, with a less reduced protein intake, Milovanova et al. included 33% of the screened patients [3]. Overall, only 30% of the pre-selected patients correctly achieved a VLPD + KAs [71]. Zoccali et al. suggested that it is unrealistic to have, at a country level, about one-third of the whole workforce of dietitians entirely dedicated to the follow-up of CKD patients [72].

Another key question is whether LPD/VLPD + KAs is applicable to the elderly patients who presently dominate the CKD population but are less likely to be recruited into RCTs. Elderly patients have already spontaneous lower dietary protein intakes and are developing sarcopenia. Cognitive deficit may also challenge an effective and safe follow-up of such restricted diets. The RCT Diet or Dialysis in Elderly (DODE) trial, which included 112 patients [73], observed that VLPD + KAs was an efficient strategy to delay the need for dialysis. The interpretation of this study is not consensual and it seems that VLPD + KAs might have increased recurring episodes of fluid overload or hyperkalemia [74]. However, the mortality rate was not different between the two groups and the hospitalization rate was higher in the dialysis group.

If some preliminary data suggest that VLPD + KAs supplementation is interesting and safe in dialysis patients, after renal transplantation or during pregnancy in CKD women, further large studies are necessary [75,76]. Lastly, the best timing, e.g., the kidney disease stage to start a KA supplementation, has not been clearly defined.

A final question is: what are the optimal level and composition of KAs that are needed? One observational study showed that a low dose of KAs had a beneficial effect to slow down renal function deterioration in pre-dialysis patients [77]. Some studies suggest that amino acid like lysine supplementation may improve vascular calcification [78]. By contrast, tryptophan and tyrosine are precursors of URS and, therefore, may be deleterious. As mentioned above, the role of BCAA supplementation must also be clarified.

## 6. Conclusion

The results of this review support the beneficial effects of VLDP + KAs in CKD patients on renal function and different metabolic parameters in particular acidosis, insulin resistance, and bone metabolism without alteration of the nutritional status. However, evidence is inconclusive regarding the effect of VLDP + KA on mortality and cardiovascular events. The recommendation for the use of KAs has not reached consensus and remains a second-choice alternative. Few international and national societies currently recommend KAs supplementation in the management of CKD (Table 4).

## 7. Future Directions

The low adherence is a main caveat when prescribing VLPD + KAs. Ethnic and cultural dietary variations between CKD populations raise the question of whether the results of protein restriction trials are applicable on an international basis. Another limitation is the side effects of KAs. They can induce hypercalcemic episodes due to their calcium content and uncomfortable gastrointestinal symptoms. In a developing country without easy access of the medical resource, as compared with the maintenance dialysis expenses, the low cost of KAs may be an attractive alternative strategy. Considering an average yearly cost of about 34,072 € for dialysis and 1440 € for the diet, treating patients with VLDP + KA and delaying dialysis treatment would allow significant financial savings [85]. In Western countries, VLPD + KAs may be proposed as a possible tool to reduce CKD costs and postpone dialysis when waiting for fistula maturation or for a living kidney transplant. The personalized medicine is one of the emerging clinical models in the new millennium. Additionally, multiple choice regarding diet linked to patient characteristics may lead to better adherence [86]. Further studies are needed to identify factors that influence patient adherence. Incremental dialysis, which combines a nutritional-dialysis approach, may be an option for avoiding the abrupt start of thrice weekly hemodialysis and should be one of the possible new treatment options. Additionally, further research will be required to establish the optimal KAs dose and role of this diet on uremic toxin production.

## Figures and Tables

**Figure 1 nutrients-11-02071-f001:**
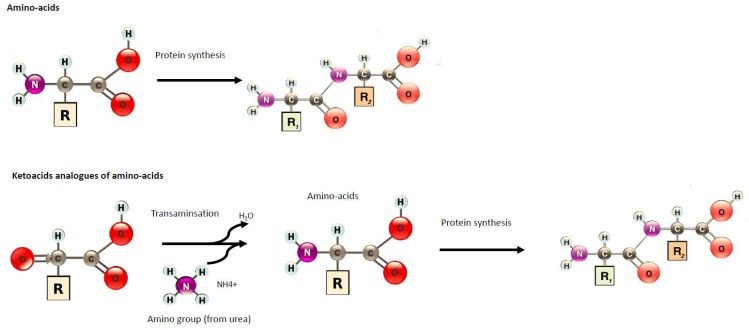
Amino-acid and transamination of ketoacid analogues of amino acids in order to synthesize protein.

**Figure 2 nutrients-11-02071-f002:**
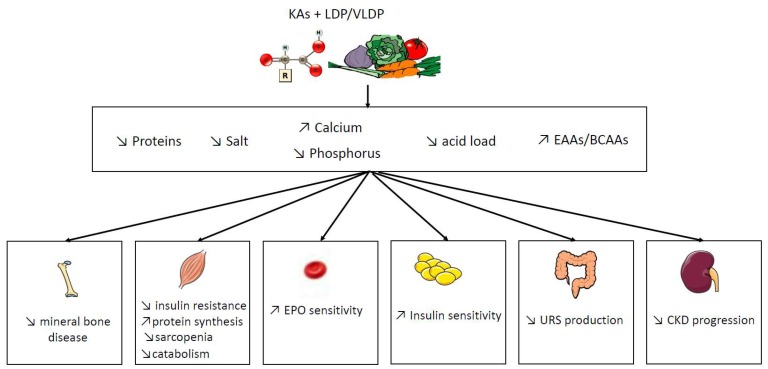
Proven and controversial mechanism of VLDP/LPD + KAs supplementation in CKD Abbreviations: URS: uremic retention solutes, EAAs: essential amino acids, BCAAs: branched-chain amino acids, LPD: low protein diet, VLDP: very low protein diet, GFR: glomerular filtration rate, and KAs: ketoacid analogues.

**Table 1 nutrients-11-02071-t001:** Ketoacid analogues composition.

Component Name	mg/pill
Ca-Keto-dl-isoleucine	67
Ca-Ketoeucine	101
Ca-Ketophénylalanine	68
Ca-Ketovaline	86
Ca-Hydroxy-dl-methionine	59
l-Lysine monoacetate	105
l-Threonine	53
l-Tryptophan	23
l-Histidine	38
l-Tyrosine	30

Ca: calcium.

**Table 2 nutrients-11-02071-t002:** Animal studies that examined the effects of VLPD/LPD supplemented with ketoacid analogues on various endpoints.

Study	Models	Diet Intervention	Follow-Up	Results (LPD vs. VLDP/LPD + KAs)
Wang et al., 2018 [8]	5/6 nephrectomy rats	NPD: 22% protein vs. LPD: 6% protein vs. LPD + KAs: 5% protein plus 1% KA	24 weeks	↓ muscle atrophy ↑ activities of mitochondrial electron transport chain complexes and mitochondrial respiration, ↓ muscle oxidative damage ↑body weight
Liu et al., 2018 [9]	KKAy mice, an early type 2 DN model	NPD: 22% protein vs. LPD: 6% protein vs. LPD + KAs: 5% protein plus 1% KA	12 weeks	↓ proteinuria ↓ mesangial proliferation and oxidative stress ↑ serum albumin and body weight No difference in creatinine and GFR
Zhang et al., 2016 [10]	3/4 nephrectomy rats	NPD: 18% protein vs. LPD: 6% protein vs. LPD + KAs: 5% protein plus 1% KA	12 weeks	↓ proteinuria ↓ intrarenal RAS activation. ↓ transforming growth factor-β1 in the mesangial cells
Zhang et al., 2015 [11]	5/6 nephrectomy rats	NPD: 11 g/kg/day protein vs. LPD: 3 g/kg/day protein vs. LPD + KAs: 3 g/kg/day protein which including 5% protein plus 1% KA	24 weeks	↑ body weight, gastrocnemius muscle mass ↓ autophagy marker in muscle No difference of inflammation markers
Wang et al., 2014 [12]	5/6 nephrectomy rats	NPD: 22% protein vs. LPD: 6% protein vs. LPD + KAs: 5% protein plus 1% KA	24 weeks	↑improved protein synthesis and increased related mediators such as phosphorylated Akt in the muscle ↓ protein degradation and proteasome activity in the muscle
Gao et al., 2010 [13]	5/6 Nephrectomy rats	NPD: 22% protein vs. LPD: 6% protein vs. LPD + KAs: 5% protein plus 1% KA	24 weeks	↓ proteinuria, glomerular sclerosis, and tubulointerstitial fibrosis ↑renal function ↑ body weight and albumin ↓ lipid and protein oxidative products
Gao et al., 2011 [14]	5/6 Nephrectomy rats	NPD: 22% protein vs. LPD: 6% protein vs. LPD + KAs: 5% protein plus 1% KA	6 months	↑ body weight and albumin ↑ Kruppel-like factor-15, a transcription factor shown to reduce fibrosis
Maniar et al., 1992 [15]	5/6 Nephrectomy rats	NPD: 16% casein vs. LPD + EAA: 6% casein + EAA vs. LPD + KAs: 6% casein + KA	3 months	No difference on body weight No difference on proteinuria vs. LDP + EAA but reduction vs. NPD ↓ creatinemia, proteinuria, glomerular sclerosis, and tubulointerstitial fibrosis vs. NPD but no difference vs. LPD + EAA ↑survival vs. NPD but no difference vs. LPD + EAA
Laouari et al., 1991 [16]	5/6 Nephrectomy rats	NPD: 12% casein vs. LPD + EAAs: 5% casein + EAA vs. LPD + KAs: 5% casein + KA		↓Appetite and growth No increase in BCAAs
Benjelloun et al., 1993 [17]	Rats with after a single 5 mg/kg intravenous injection of Adriamycin: a model of induces glomerular damage in glomerulonephritis.	NPD: 21% protein vs. LPD + KAs: 6% protein plus KA	15 days	↓ proteinuria ↓ glycosaminoglycan excretion and glomerular glycosaminoglycan contents
Barsotti et al; 1988 [18]	5/6 Nephrectomy rats	NPD: 20.5% protein vs. LPD + KAs: 3.3% protein plus 7.5% KA	3 months	↑survival ↑ GFR ↓ proteinuria and histological damage of kidney No difference in body weight and albuminuria
Meisinger et al., 1987 [19]	5/6 Nephrectomy rats	LPD: 8% protein vs. LPD + KAs: 8% protein plus KA	3 months	↓ proteinuria

NPD: normal protein diet. HPD: high protein diet. GFR: estimated Glomerular Filtration Rate. LPD: Low protein diet. KAs: ketoacid analogues. EAAs: essential amino acids. BCAAs: branched-chain amino acids; RAS: renin angiotensin system; NPD: normal protein diet.

**Table 3 nutrients-11-02071-t003:** Main RCTs that examined the effects of LPD or VLDP/LPD supplemented with ketoacid analogues on various endpoints in non-dialysis patients with eDFG under 60 mL/min/1.73 m^2^.

Study	Design of Study	Diet	Follow-Up	Results	Comments
Milovanova et al., 2018 [3]	RCT *n* = 42 in LPD + KA vs. LPD *n* = 37 Non-diabetic CKD 3B–4	LPD (0.6 g/kg of body weight/day, comprising 0.3 g of vegetable protein and 0.3 g of animal protein, phosphorus content ≤ 800 mg/day and calories: 34–35 kcal/kg/day) vs. LPD + KA: 0.6 g/kg of body weight/day	14 months	↑ eGFR (29.1 L/min/1.73 m^2^ vs. 26.6) ↓SBP ↑BMI and muscle body mass NO change in albumin levels No change in lipids parameters ↓ phosphate, FGF23, and PTH levels ↑Klotho levels and phosphate binder uses ↑bicarbonates levels	Similar protein intake in both group Long follow up
Di Iorio et al., 2018 [20]	RCT, crossover trial CKD stages 3B–4 Group A1: 3 months of FD, 6 months of VLPD + KA, 3 months of FD and 6 months of MD Group B: 3 months of FD, 6 months of MD, 3 months of FD and 6 months of VLPD + KA. *n* = 30 in each group	FD: proteins 1 g/kg body weight (bw)/day (animal proteins 50–70 g/day, vegetal proteins 15–20 g/day), energy 30–35 kcal/bw/day, calcium (Ca) 1.1–1.3 g/day, phosphorus (P) 1.2–1.5 g/day, sodium (Na) 6 g/day and potassium (K) 2–4 g/day. MD: proteins 0.7–0.8 g/kg bw/day (animal proteins 30–40 g/day, vegetal proteins 40–50 g/day), energy 30–35 kcal/bw/day, Ca 1.1–1.3 g/day, P 1.2–1.5 g/day, Na 2.5–3 g/day and K 2–4 g/day. VLPD + KA: proteins 0.3–0.5 g/kg bw/day (animal proteins 0 g/day, vegetal proteins 30–40 g/day), energy 30–35 kcal/bw/day, Ca 1.1–1.3 g/day, P 0.6–0.8 g/day, Na 6 g/day, K 2–4 g/day plus a mixture of KA	6 months	↓ SBP No change in creatinuria ↓proteinuria ↓ phosphate, FGF23, and PTH levels ↑bicarbonates levels ↑Hg levels ↓protein carbamylation	Sodium intake and phosphore intake was reduce in VLDP + KA group
Garneata et al., 2016 [21]	RCT CKD stage 4–5, proteinuria < 1 g/24 h *n* = 207	LPD = 0.6 g protein/kg per day vs. VLPD + KA = vegetarian diet, 0.3 g protein/kg per day + KA	15 months	↓ RRT initiation or a >50% reduction in the initial GFR (13% in KA+LDP vs. 42% in LPD reached the primary composite efficacy point i.e., RRT initiation or a >50% reduction in the initial GFR) ↓CRP ↑bicarbonates levels ↓uric acid ↓ phosphate, FGF23 and PTH levels and phosphate binder uses No difference in proteinuria No difference of death and CV events No difference of albumin, BMI No change in lipids parameters	Long follow up Large effective Only 14% of patients screened was included
Di Iorio et al., 2012 [22]	RCT, crossover trial eGFR < 55 and > 20 mL/min/1.73 m^2^ Group A: VLDP + KA during the first week and LPD during the second week Group B: LPD during the first week and a VLPD + KA during the second week. *n* = 16 in each group	LPD = 0.6 g protein/kg per day vs. VLPD + KA = 0.3 g protein/kg per day + KA	1 week	↓ phosphate (−12%), FGF23 (−33.5) No change on calcium a post hoc of this study, ↓ indoxyl sulfate [23] ↑bicarbonates levels	Short exposition
Di Iorio et al., 2009 [24]	RCT, crossover trial eGFR < 55 and > 20 mL/min Group A: VLDP + KA during 6 month and a LPD during 6 month Group B: LPD during 6 month and a VLDP + KA during 6 month. *n* = 16 in each group 32 patients	LPD = 0.6 g protein/kg per day vs. VLPD + KA = 0.3 g protein/kg per day + KA	6 months	↓proteinuria and AGE	Open label Phosphor intake was different and lower in VLDP+ KA
Menon et al., 2009 [25]	Post hoc study of MDRD study B CKD stage 4 nondiabetic *n* = 255	LPD = 0.6 g protein/kg per day vs. VLPD + KA = 0.3 g protein/kg per day + KA	10.2 years	No delay progression to kidney failure ↑the risk of death.	Long follow up without intervention -Observance and protein intake was not monitored during the follow up
Teplan et al., 2008 [4]	RCT, double-blind placebo CKD stage 4 *n* = 111	LDP: 0.6 g protein/kg per day vs. LPD + KA: 0.6 g protein/kg per day + KA	36 months	↓ADMA ↓ BMI and visceral body fat in obese patients ↓proteinuria ↓ glycated hemoglobin ↓LDL-cholesterol	Mean BMI was > 30 kg/m^2^ at the inclusion Long follow up No difference of protein intake Using a placebo
Mircescu et al., 2007 [26]	RCT eGFR <30 mL/min/1.73 m^2^, nondiabetic *n* = 53	VLPD + KA =0.3 g/kg vegetable proteins + KA vs. LPD =0.6 g/kg/d)	48 weeks	↑bicarbonates levels ↑calcium levels and ↓ phosphate lower percentages of patients in group I required renal replacement therapy initiation (4% vs. 27%). No change of rate of eGFR and proteinuria No change in SBP	Open label
Gennari et al., 2006 [27]	Post hoc study of MDRD study RCT CKD stage 4–5 *n* = 255	LPD = 0.6 g protein/kg per day vs. VLPD + KA = 0.3 g protein/kg per day + KA	2,2 years	No significant effect of diet on serum total CO2 was seen	
Menon et al., 2005 [28]	Post oc study of MDRD study RCT CKD stage 4–5 *n* = 255	LPD = 0.6 g protein/kg per day vs. VLPD + KA = 0.3 g protein/kg per day + KA	2.2 years	↓ homocysteinemia by 24% at 1 year	
Feiten et al., 2005 [29]	RCT *n* = 24 eGFR <25 mL/min	VLPD + KA = 0.3 g/kg vegetable proteins + KA vs. LPD = 0.6 g/kg/d	4 months	↑bicarbonates levels No change on calcium levels ↓ phosphate and PTH Decrease the progression of renal decline function of rate of eGFR No change in lipid parameters No change in nutritional status (BMI, albumin)	Open label Short time of follow up Significant reduction in dietary phosphorus (529 ± 109 to 373 ± 125 mg/day, *p* < 0.05)
Prakash et al., 2004 [30]	RCT, double-blind placebo eGFR:28 mL/min/1.73 m^2^ *n* = 34	LPD = 0.6 g protein/kg per day + placebo vs. VLPD = 0.3 g protein/kg per day + KA	9 months	preserve mGFR (−2% in LDP + KA vs. −21% in LPD) No effect on proteinuria No effect of BMI and albumin	Measure of GFR with 99mTc-DTPA The placebo is problematic because protein intake was different between both groups.
Teplan et al., 2003 [5]	RCT eGFR: 22–36 mL/min/1.73 m^2^ *n* = 186	LPD 0.6 g protein/kg per day + rhuEPO + KA vs. LPD: 0.6 g protein/kg per day + rhuEPO vs. LPD: 0.6 g protein/kg per day	3 years	Slower progression of CKD ↓proteinuria ↓LDL-cholesterol No change in SBP ↑albumin ↑ plasmatic leucine levels	Role of rhuEPO unclear Insulin clearance
Di Iorio et al., 2003 [31]	RCT eGFR: < or =25 mL/min/1.73 m^2^ *n* = 10 in each group	LPD = 0.6 g protein/kg per day vs. VLPD = 0.3 g protein/kg per day + KA	2 years	No difference on hemoglobin ↓ EPO dose ↓ phosphate and PTH No change in BMI and albumin No difference in the rate of RRT initiation (8 vs. 7) Slower rate of GFR decline (creatinine clearance) ↓SBP and 24 h NA excretion ↓LDL-cholesterol	Very few populations
Bernhard et al., 2001 [6]	RCT CKD stage 4–5 *n* = 6 in each group	LPD = 0.6 g protein/kg per day vs. LPD + KA = 0.6 g protein/kg per day + KA	3 months	No difference could be attributed to the ketoanalogs total body flux and leucine oxidation No difference on phosphorus, calcium levels No difference on BMI and albumin No difference in renal function and proteinuria No difference on bicarbonatemia	KA is metabolically safe Short follow-up Small effective
Malvy et al., 1999 [32]	RCT eGFR<20 mL/min/1.73 m^2^ *n* = 50	LPD:LPD = 0.65 g protein/kg per day + Ca+ vs. VLPD + KA = 0.3 g protein/kg per day + KA	3 months or time to eGFR < 5 mL/min/1.73 m^2^ or RRT	No difference on GFR progression ↑calcium levels ↓ phosphate and PTH No difference on lipid parameters	
Kopple et al., 1997 [33]	Post hoc study of MDRD study RCT CKD stage 4–5 *n* = 255	LPD = 0.6 g protein/kg per day vs. VLPD + KA = 0.3 g protein/kg per day + KA	2,2 years	No difference of death and first hospitalization ↑ albumin ↓ transferrin, body wt, percent body fat, arm muscle area, and urine creatinine excretion No correlation between nutritional parameters and death or hospitalization ↓ energy intake	
Levey et al., 1996 [34]	Post hoc study of MDRD study RCT CKD stage 4–5 *n* = 255	LPD = 0.6 g protein/kg per day vs. VLPD + KA = 0.3 g protein/kg per day + KA	2.2 years	A 0.2 g/kg/d lower achieved total protein intake was associated with a 1.15 mL/min/yr slower mean decline in GFR (*p* = 0.011), which is equivalent to 29% of the mean GFR decline	Reanalyze of MDRD study by using correlations of protein intake with a rate of decline in GFR and time to renal failure
Klahr et al., 1994 Study 2 [35]	RCT CKD stage 4–5 *n* = 255	LPD = 0.6 g protein/kg per day vs. VLPD + KA = 0.3 g protein/kg per day + KA	27 months	Marginally slower eGFR decline (−19% in LPD vs. 12% in VLDP + KA, *p* 0.067) No significant interactions between blood-pressure interventions and the rate of decline in eGFR No difference on albumin No difference in proteinuria	-Large RCT study -Good adherence of diet -Measured GFR with iothalamate
Coggins et al. 1994 [36]	Feasibility phase of the MDRD Study eGFR: 8 to 56 mL/min/1.73 m^2^ *n* = 96 25 participants were excluded	LPD = 0.6 g protein/kg per day vs. VLPD + KA = 0.3 g protein/kg per day + KA	6 months	No difference on lipid parameters	Pilot study
Lindenau et al. 1990 [37]	RCT eGFR<15 mL/min/1.73 m^2^ *n* = 40	LPD = 0.6 g protein/kg per day + Ca+ vs. VLPD + KA = 0.4 g protein/kg per day + KA	12 months	Improvement in osteo-fibrotic as well as in osteo-malacic changes	A calcium supplementation was given in LPD diet as a control for KA
Jungers et al. 1987 [38]	RCT CKD stage 5 *n* = 19	LPD = 0.6 g protein/kg per day + Ca+ vs. VLPD + KA = 0.4 g protein/kg per day + KA	12 months	No difference on biochemical or morphometric sign of de-nutrition ↑mean renal survival duration until dialysis	Small and effective
Hecking et al., 1982 [7]	RCT Mean eGFR: 10.8 mL/min/1.73 m^2^ *n* = 15	LPD = 0.6 g protein/kg per day + Ca+ vs. LPD + KA = 0.6 g protein/kg per day + KA or EAA or placebo	3 weeks per periods	↓ phosphate No difference on GFR and proteinuria No difference on lipids parameters No difference on albumin	Small and effective versus the placebo

FD: Free diet. P: phosphorus. MDRD: Modification of Diet in the Renal Disease Study. eGFR: estimated Glomerular Filtration Rate. RRT: renal replacement therapy. FGF23: Fibroblast Growth Factor 23. LPD: Low protein diet. VLDP: Very low protein diet. KA: Keto-analogues. RCT: randomized controlled trial. EAA: essential amino acids; PTH: parathyroid hormone.

**Table 4 nutrients-11-02071-t004:** VLDP/LDP and KA in non-dialyzed patients with chronic kidney disease.

CKD Stage	NKF/KDOQI Clinical Practice Guidelines for Nutrition in Chronic Renal Failure, 2000 [79]	ESPEN Guidelines on Enteral Nutrition: Adult Renal Failure, 2006 [80]	Australian KHA-CARI Guidelines [81]	KDIGO, 2012 [82]	International Society of Renal Nutrition and Metabolism, 2013 [83]	Review, 2017, NEJM [84]
3	0.60–0.75 g/kg/day of protein	0.55–0.6 g/kg/day of protein	0.75–1.0 g/kg/day of protein	<1.3 g/kg/day protein	0.6 and 0.8 g/kg/day protein, ≥50% of protein of HBV	<1.0 g/kg/day protein (consider 0.6–0.8 if eGFR <45 mL/min/1.73 m^2^ or rapid progression)
4–5	0.60–0.75 g/kg/day of protein	0.55–0.6 g/kg/day of protein (2/3 HBV) or ~0.3 g/kg/day of protein supplemented with KAs/EAAs (0.1 g/kg/day)	0.75–1.0 g/kg/day	0.8 g/kg/day protein	0.6 and 0.8 g/kg/day protein, ≥50% of protein of HBV	0.6–0.8 g/kg/day protein, including 50% HBV protein, or <0.6 with addition of EAAs or KAs (0.1 g/kg/day)

HBV: high biological value. KAs: ketoacids analogues. EAAs: essential amino acids.

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
