# Peer review of "Ketoacid Analogues Supplementation in Chronic Kidney Disease and Future Perspectives"

_nutrients, 2019, doi:10.3390/nu11092071_

Round 1
Reviewer 1 Report
In the revised manuscript, authors have addressed the reviewer's concerns. The reviewer has no more comments.
Author Response
Thank you
Reviewer 2 Report
Much of this review is a narrative and not set up as a typical systematic review. I recommend some restructuring of the paper and removing extraneous sections such as the case study. The first sections (particularly the first 4 pages) of the review require the most work. Starting at section 3.1, there are mostly only minor edits to be made - an interesting and insightful summary of the literature. Please find the track changes version in the attachment.
The abstract does not adequately describe the contents of the manuscript. Its focus of the one study does not match the focus of your systematic review. I would recommend instead describing the methods and major findings of your review.
Case studies are at a very low level of evidence and thus, its description/inclusion simply does not belong in a review of the current literature on this topic.
Methods of the systematic review need to precede mention of the tables.
You state the review is 'without language restriction' but then only include clinical studies that were available in English.
Was ESRD or renal used as search terms as these were very common in the past. Was CKD spelled out and searched. Have some papers been missed due to the limited search terms ?
The conclusion mentions important considerations but seems to belong just prior to the conclusion – perhaps with a heading ‘Future directions » The conclusion should instead be focused on the the most important findings of your review.

Author Response
Reviewer 2
1-Much of this review is a narrative and not set up as a typical systematic review. I recommend some restructuring of the paper and removing extraneous sections such as the case study. The first sections (particularly the first 4 pages) of the review require the most work. Starting at section 3.1, there are mostly only minor edits to be made - an interesting and insightful summary of the literature. Please find the track changes version in the attachment.
We have reduced the first part of the article and performed some restructuring like reducing the introduction, suppressing part 2, removing the case study.
2-The abstract does not adequately describe the contents of the manuscript. Its focus of the one study does not match the focus of your systematic review. I would recommend instead describing the methods and major findings of your review.
We have modified the abstract.
3-Case studies are at a very low level of evidence and thus, its description/inclusion simply does not belong in a review of the current literature on this topic.
As suggested by the reviewer, we have removed the case study.
4-Methods of the systematic review need to precede mention of the tables.
We added a specific part about methods of the systematic review (see 3.methods) before mention of the tables.
5-You state the review is 'without language restriction' but then only include clinical studies that were available in English.
For the literature research, we didn’t perform a language restriction initially. But we have included at the end only the articles with full text available in English. We have clarified this point. (See 3.Methods)
6-Was ESRD or renal used as search terms as these were very common in the past. Was CKD spelled out and searched. Have some papers been missed due to the limited search terms?
We have not included initially ESRD because we are focused only on patient without dialysis. We have used CKD or chronic kidney disease as search terms. We have performed a new research including renal, ESRD or end stage renal disease or kidney or CKD or chronic kidney disease and ketoacids or ketoanalogs or very low protein diet. We found 1612 additional articles: 11084 papers in Medline, 1105 in WEB of SCIENCES, 823 in SCOPUS, and 120 in CENTRAL. On summary, we have identified 13 132 potentially relevant references from the search (12453 records without duplicates; Supplementary Fig. 1). After title and abstract screening, 140 full-text articles were considered for inclusion. Twenty-three full-text studies were selected in order to analyze the potential effects of KAs.
In conclusion, we didn’t find new articles with using ESRD or renal as search terms.
We have modified the manuscript according to the reviewer’s suggestions. See methods 3 and supplementary Figure 1.
7-The conclusion mentions important considerations but seems to belong just prior to the conclusion – perhaps with a heading ‘Future directions » The conclusion should instead be focused on the the most important findings of your review.
We have modified the conclusion. We have added a part “future directions” and we have added a short conclusion that highlighted the most important findings of our systematic review.
Round 2
Reviewer 2 Report
Abstract revisions are required. The abstract needs to appropriately reflect the content of the manuscript. Instead, it contains basic knowledge statements in the background and results that do not reflect comprehensively, the review.
Thank you for removing the case study. However, the introduction requires further revision. For example..."Brenner et al. [1] showed that dietary protein restriction slowed progression of chronic kidney disease (CKD) through renal hemodynamic improvements and suggested that protein restriction could limit CKD progression. Therefore, the aims of a nutritional intervention during CKD have two main objectives: 1) minimize uremic toxicity and delay the need to start dialysis and 2) avoid cachexia." The second sentence does not logically follow the first. Please revise the first 3 paragraphs of the review to logical progress to the aim.
As is, the manuscript reports a systematic "search" of the literature and a "narrative" review that is interesting and well done, but is not clearly a systematic review of the literature searched and presented in the tables. Without adherence to the PRISMA guidelines, "systematic" needs to be removed from the title and elsewhere to reflect the review's narrative nature. If the goal is a systematic review, please see the PRISMA guidelines and checklist for reporting, make the needed revisions. Please also indicate why a meta-analysis was not carried out.
Author Response
1-Abstract revisions are required. The abstract needs to appropriately reflect the content of the manuscript. Instead, it contains basic knowledge statements in the background and results that do not reflect comprehensively, the review.
We have changed the abstract, we have tried to summarize the key points of our review.
2-Thank you for removing the case study. However, the introduction requires further revision. For example..."Brenner et al. [1] showed that dietary protein restriction slowed progression of chronic kidney disease (CKD) through renal hemodynamic improvements and suggested that protein restriction could limit CKD progression. Therefore, the aims of a nutritional intervention during CKD have two main objectives: 1) minimize uremic toxicity and delay the need to start dialysis and 2) avoid cachexia." The second sentence does not logically follow the first. Please revise the first 3 paragraphs of the review to logical progress to the aim.
We have revised the first 3 paragraphs.
3-As is, the manuscript reports a systematic "search" of the literature and a "narrative" review that is interesting and well done, but is not clearly a systematic review of the literature searched and presented in the tables. Without adherence to the PRISMA guidelines, "systematic" needs to be removed from the title and elsewhere to reflect the review's narrative nature. If the goal is a systematic review, please see the PRISMA guidelines and checklist for reporting, make the needed revisions. Please also indicate why a meta-analysis was not carried out
Thank you for these suggestions. You are right, it is not a systematic review with the PRISMA guidelines. We have suppressed the term “systematic” in the title and in the manuscript. The goal is more a narrative review in order to give a up to date of knowledge about KAs supplementation and propose future directions. We have proposed an exhaustive literature review with selecting experimental and randomized clinical studies. We have chosen to not performed a meta-analysis because recently different meta-analysis have been performed (Li A et al, Nutrients, 2091, Hahn et al Cochrane Database Syst Rev. 2018). Also, the number and quality of studies about KAs on uremic toxins and anabolic action are not sufficient to carry out a meta-analysis on these points.
Round 3
Reviewer 2 Report
The attached document has some minor edits.

Author Response
Thank you for your corrections. I did the modifications.

This manuscript is a resubmission of an earlier submission. The following is a list of the peer review reports and author responses from that submission.
Round 1
Reviewer 1 Report
In this manuscript, authors elegantly reviewed the benefits of supplementation by nitrogen-free ketoacid analogues (KA) in chronic kidney disease (CKD) patients under low protein diets (LPD) and very low protein diets (VLPD) not only to reduce malnutrition and uremic toxin production but also to prevent the progression of kidney injury. The subject seems to be currently important, and the contents are easy to understand. Although this manuscript does not need major revision, the reviewer has some comments and questions that should be addressed in revised paper, as following;
1. In Introduction, authors should explain how they searched for and chose reference articles that were included in the text and tables in order to ensure unbiased selection of articles.
2. In Figure 2, did authors cite the case from a previous report? If so, add the reference. Otherwise, authors should explain the case in more detail. More information including the patient characteristics, renal function, and does of KA and protein intake, should be described.
3. Are there any evidence showing that supplementation of KA alone without low protein diets has any benefits on prevention of the progression of CKD?
4. The reviewer thinks that the part of conclusion is too long. Figure 3 should be transfer to the end of “2-Potential Benefit of Ketoacid Analogues”.
5. There are some spelling errors such as “EEA” in the text. Please correct them.
Reviewer 2 Report
In present study by Koppe et al reviewed recent studies on Ketoacid analogues supplementation in chronic kidney disease. Author has described historical perspective, basic biochemistry, and future perspective of KAs. They have extensively extracted data/information from many animal and human studies. Author did an excellent job on that. It highly demand and huge controversial topic/field as there is not much molecular evidence that support the beneficial effect of KAs in CKD or any other kidney complications.
a) What is the way author has searched the database? What database and key word used for this study?
b) Abstract
1. Change kidney patients to CKD or kidney disease patients.
2. Why does author think that largest randomized trial of VLDP+KAs showed no beneficial effect but the practice should be included in the clinical recommendations?
c) Introduction
1. Please include reference for Giordano and Walser reports.
2. Overall introduction is confusing or not enough rational given for the review.
d) WHAT ARE KETOACID ANALOGUES?
1. Move… KAs are precursor of ….to… beginning of the paragraph and start the relevance of CKD to KAs.
2. Table 1: Mg=mega gram and mg = milligram. Most likely author referring to milligram. Please change if so.
e) POTENTIAL BENEFIT OF KETOACID ANALOGUES
1. Table 4: what Y-Axis represents? Which patient’s samples were used for this figure?
2. What is the reason of discussing about cachexia in here?
f) This review is much similar and recent version to Shah A et al in AJKD, 2015 “Shah A et al, Is there a role for ketoacid supplements in the management of CKD?, AJKD,2015. Author should consider citing this paper.”
g) Potential limitations should be included as there are side effects KA-EAA supplementations. And also, how economically feasible this diet?
h) Overall manuscript is too long with repeating fact in different section and not much additional information added. In many places author used incomplete sentences and past participle instead of past tense. It Would improve the manuscript’s purpose if language is clear.
Reviewer 3 Report
Thank you for the opportunity to review. Unfortunately there is little new to report in this manuscript. The information contained has been reported elsewhere in many venues. The MDRD study, which the authors have repeated eluded to, was severely flawed in recruitment bias when a large number of polycystic disease patients were included in the sample. Their progression in CKD was different than non-polycystic. This fact is well known and published. Second, Table 2 already shows keto-acids are incorporated into all the major clinical guidelines reported globally. The authors acknowledge it is recommended as a second line approach, which I concur. The cost and adherence barriers are well known, particularly in westernized populations. Third, Table 3 of animal studies is not applicable. In nephrology, we are far beyond animal studies for safety. The relevance is questionable for your journal reader population. Fourth, all the authors do identify risks of malnutrition and protein wasting - the use of KA cannot become mainstream as they elude. Additional RCT will not promote its use as a first line agent for the cost and adherence issues that the authors state. The case study of a single patient was informative, but too little information was given. CKD patients are highly variable in their clinical conditions and effects. If the authors want to contribute to the literature, expand that case study with clear and focused information on how the KA was clinical relevant for this patient. The manuscript, although thorough in scope, provides nothing new to the literature.